# Challenges for Porcine Reproductive and Respiratory Syndrome (PRRS) Vaccine Design: Reviewing Virus Glycoprotein Interactions with CD163 and Targets of Virus Neutralization

**DOI:** 10.3390/vetsci6010009

**Published:** 2019-01-17

**Authors:** Ana M. M. Stoian, Raymond R. R. Rowland

**Affiliations:** Department of Diagnostic Medicine and Pathobiology, Kansas State University, Manhattan, KS 66506, USA; astoian@vet.k-state.edu

**Keywords:** porcine reproductive and respiratory syndrome virus, PRRSV, CD163, GP4, GP5, neutralizing antibody

## Abstract

One of the main participants associated with the onset and maintenance of the porcine respiratory disease complex (PRDC) syndrome is porcine reproductive and respiratory syndrome virus (PRRSV), an RNA virus that has plagued the swine industry for 30 years. The development of effective PRRS vaccines, which deviate from live virus designs, would be an important step towards the control of PRRS. Potential vaccine antigens are found in the five surface proteins of the virus, which form covalent and multiple noncovalent interactions and possess hypervariable epitopes. Consequences of this complex surface structure include antigenic variability and escape from immunity, thus presenting challenges in the development of new vaccines capable of generating broadly sterilizing immunity. One potential vaccine target is the induction of antibody that disrupts the interaction between the macrophage CD163 receptor and the GP2, GP3, and GP4 heterotrimer that protrudes from the surface of the virion. Studies to understand this interaction by mapping mutations that appear following the escape of virus from neutralizing antibody identify the ectodomain regions of GP5 and M as important immune sites. As a target for antibody, GP5 possesses a conserved epitope flanked by N-glycosylation sites and hypervariable regions, a pattern of conserved epitopes shared by other viruses. Resolving this apparent conundrum is needed to advance PRRS vaccine development.

## 1. Introduction

Porcine reproductive and respiratory syndrome virus (PRRSV) consists of two species: PRRSV-1 isolates are of European origin while PRRSV-2 originated in North America. PRRSV-1 and PRRSV-2 share ~60% identity at the nucleotide level. In addition, each species can be further divided into several clades or substrains. PRRSV, along with the closely related mouse virus, lactate dehydrogenase-elevating virus (LDV), are part of the Arteriviridae family, Variarterivirinae subfamily [1]. Further on, the two PRRSV species have been placed in genus Betaarterivirus, with Ampobartevirus and Eurpobartevirus as corresponding subgenuses. Subsequently, *Betaarterivirus suid 1* species was created for the original PRRSV-1 and *Betaarterivirus suid 2* species for the PRRSV-2, respectively. The progenitor virus for PRRSV-1 likely emerged in Eastern Europe and Russia followed by the introduction of PRRSV into North America, perhaps through the introduction of Russian wild boar [2]. A separate course of evolution in North America produced PRRSV-2. Perhaps the most remarkable aspect of PRRS virus evolution is the simultaneous emergence of PRRSV-1 and PRRSV-2, which produce similar disease signs and possess a similar epidemiology/ecology. Therefore, PRRSV is a good example of how a virus with unique biological properties is able to effectively exploit unique ecological niches created by a modern swine industry.

The PRRSV genome possesses at least ten open reading frames (ORFs) flanked by a 5′ leader and 3′ untranslated region followed by poly-A tail. The nonstructural proteins, encoded by ORF1a and ORF1b, possess protease, replicase and host gene modulation functions. The 3′ end of the genome codes for at least eight structural proteins translated from a nested 3’-coterminal set of subgenomic mRNAs possessing a common leader, a hallmark feature of the genus and the Nidovirus order. The major structural proteins, GP5, matrix (M), and nucleocapsid (N) are encoded by ORFs 5, 6, and 7, respectively. GP5 and M generally exist as a GP5-M heterodimer; however, GP5 homodimers have been identified [3]. GP2, GP3, and GP4 are minor surface glycoproteins (GPs) derived from ORFs 2, 3, and 4, respectively. Two very small nonglycosylated proteins—2b (or E) and 5a—are translated from ORF2b and ORF5a, respectively [4,5]. In 2013, Kappes et al. [6] described the association of the nonstructural protein, nsp2, with the virion. However, there are no published data demonstrating that anti-nsp2 antibodies possess neutralizing activity.

The topological features of the virion surface are described in Spilman et al. [7], who performed cryo-electron microscopy followed by tomographic reconstruction of purified virions derived from MARC-145 cells infected with a PRRSV-2 isolate. The surface of the virion is smooth, reflecting the predominance of the short peptide sequences formed by the ectodomains of M and GP5. A small number of protrusions rise above the surface, formed by the large ectodomains of GP2, 3, and 4. The ectodomain regions of surface proteins are illustrated in Figure 1.

The targets for PRRSV infection are cells of monocyte/macrophage origin. It is this interaction between the virus and macrophage that is responsible for respiratory distress and immune modulation, which are associated with the onset of PRDC. Van Breedam et al. [10] were the first to propose a detailed model describing how PRRSV interacts with, and then enters, the macrophage host. In this model, PRRSV infection occurs in three steps, which incorporate interactions between PRRSV and three different receptor molecules on the macrophage cell surface. The first step is the initial interaction between the PRRSV M protein and heparin sulfate (HS) [11]. Blocking or removing HS does not completely abrogate infection, suggesting that the virion–HS interaction is relatively nonspecific and is designed to bring the virion in closer proximity to the macrophage surface. The second step is a higher affinity interaction between the glycosyl residues on GP5 and sialoadhesin (Sn) on the macrophage. Support for Sn as a PRRSV receptor was initially based on the characterization of monoclonal antibodies (mAbs) prepared against macrophage antigens, which inhibited PRRSV infection. Sn was identified as the ligand for one of the mAbs. Further support for Sn comes from making PRRSV nonpermissive cells permissive for infection after transfection with a plasmid expressing a porcine Sn cDNA [12]. However, the acquisition of Sn by nonpermissive cells did not result in productive infection. Therefore, the role of Sn is to bind PRRSV to the macrophage surface; nonetheless, this interaction is not sufficient for virus internalization. The third step for infection is internalization, uncoating of the virion, and the release of virus nucleic acid in the cytoplasm. This step occurs through the interaction of the virion with CD163, a PRRSV receptor first described by Calvert et al. [13]. CD163 was identified based on the screening of cDNAs from a macrophage library that rendered PRRSV permissiveness to a nonpermissive cell line. As indicated in the model of Van Breedam et al. [10], CD163 is not an attachment factor or surface receptor, but participates in the later stages of internalization and release of the viral genome into the cytoplasm.

The role of Sn as the primary PRRSV receptor protein was first called into question by Welch and Calvert [14], who observed that the transfection of nonpermissive PK-15 cells, which lack Sn, with a plasmid vector expressing a CD163 cDNA was sufficient to render cells permissive for PRRSV infection. Furthermore, pigs possessing a knockout of the *SIGLEC-1* gene did not express Sn on macrophages, but retained the ability to support PRRSV infection to the same level as Sn wild-type pigs [15]. One explanation for how *SIGLEC-1* knockout pigs can support infection is based on the presence of alternative SIGLEC proteins, such as SIGLEC-10, which can substitute for the SIGLEC-1 protein [16].

## 2. CD163 as the Sole Receptor for PRRSV

Even though there exists a large body of work and several convincing arguments in support of multiple PRRSV proteins interacting with multiple receptors and coreceptors on macrophages and other pig cells, the overwhelming evidence supports the hypothesis that surface expression of CD163 is necessary and sufficient to support PRRSV infection. CD163 belongs to a family of proteins possessing multiple scavenger receptor cysteine-rich (SRCR) domains. Each SRCR domain consists of approximately 100 amino acids in length and possesses multiple cysteines, which form several internal disulfide bonds. A representation of porcine CD163 (pCD163) is shown in Figure 2. In addition to the nine SRCR domains, pCD163 possesses two proline-serine-threonine-rich (PST) domains. In 2016, Whitworth et al. [17] showed that pigs possessing a knockout (KO) of the *CD163* gene were nonpermissive for infection with a PRRSV-2 isolate. For these experiments, both CD163 KO and WT pigs were comingled after infection, which allowed for the continuous exposure of the CD163 KO pigs to virus shed by the infected WT pigs. In a subsequent study, macrophages from CD163 KO pigs were shown to be resistant to infection with PRRSV-1 and PRRSV-2 [18].

Resistance to PRRSV infection locates to a single CD163 domain, SRCR5. The deletion of SRCR5 in human embryonic kidney (HEK) cells transfected with CD163 cDNA mutants [19], or pigs lacking exon 7 in the *pCD163* gene, are resistant to PRRSV-1 infection [22]. Differences between PRRSV-1 and PRRSV-2 isolates in the recognition of CD163 also localize to SRCR5. Macrophages from pigs possessing a domain swap between pCD163 SRCR5 and an SRCR8 domain homolog from the human CD163-like protein retain permissiveness to PRRSV-2, but are nonpermissive to infection with PRRSV-1 strains. This difference in recognition by PRRSV-1 and PRRSV-2 is located within the 31 amino acid difference between human CD163-like SRCR8 and pCD163 SRCR5 (Figure 2). To date, the only available information on specific amino acid residues involved in the recognition of SRCR5 by PRRSV is found in Ma et al. [21], who prepared six pCD163 SRCR5 variants possessing substitutions of arginine or aspartic acid residues with alanine residues, located along the SRCR5 polypeptide. A single arginine to alanine substitution at position 561 in pCD163 SRCR5 was sufficient to reduce infection of transfected PK-15 cells. However, the requirement of R561 for PRRSV infection is unlikely, since the human CD163-like SRCR8 domain possesses a naturally occurring histidine substitution at the same 561 position (Figure 2).

Van Gorp et al. [19] showed that other pCD163 domains, besides SRCR5, participate in PRRSV infection. For these experiments, HEK cells were transfected with pCDNA 3.1 expression plasmids possessing deletions in pCD163 or domain swaps with the human CD163-like protein. Deletion of PST2 or SRCR 7–9 resulted in complete loss of infection. Deletion of PST1 produced a significant reduction in infection (Figure 2).

## 3. The Interaction between pCD163 and PRRSV Glycoproteins

The corresponding PRRSV proteins involved in interacting with CD163 are described in Das et al. [23]. For these experiments, plasmids containing ORF2 (GP2), ORF3 (GP3), ORF4 (GP4), or ORF5 (GP5) cDNAs were transfected singly or in combination into nonpermissive BHK-21 cells. High levels of PRRSV gene expression were attained by transfecting cells with an expression plasmid that placed the PRRSV glycoprotein gene under the control of a T7 RNA polymerase promoter, followed by infection of transfected cells with a vaccinia virus expressing T7 polymerase. Immunoprecipitation (IP) was performed using monospecific polyclonal serum or an antibody against flag-tagged PRRSV glycoproteins. As predicted, interactions were observed between GP2, GP3 and GP4. Interestingly, GP4 coprecipitated with GP5. BHK-21 cells were made permissive for PRRSV infection after transfection with a pCD163 cDNA plasmid. The BHK-CD163 cells were transfected with the different PRRSV GP plasmids and IP performed using an anti-CD163 antibody. Only GP2 and GP4 formed specific interactions with CD163. Together, these data confirm that GP2, 3 and 4 form a heterotrimer and that the interaction with CD163 occurs through GP2 and 4. It is possible that GP2 and 4 form multiple interactions between different domains in CD163.

As illustrated in Figure 1, GP2–4 possess multiple glycosylation sites. Das et al. [8] showed that all of the predicted N-glycosylation sites shown in Figure 1 are functional, but the N-sites are glycosylated with different efficiencies. Figure 1 also identifies those glycosylation sites on GP2, 3, 4, and 5 required for PRRSV infection. These observations are based on the infection of cells with an infectious clone possessing N-glycosylation site mutations. For example, an FL12 infections clone possessing a GP2 without the N184 glycosylation site was unable to propagate in MARC-145 cells after electroporation of the infectious clone RNA. In contrast, the removal of the N178 site had no effect on replication. For GP4, all four glycosylation sites are conserved between PRRSV-1 and PRRSV-2 viruses (Figure 3). However, the deletion of individual glycosylation sites in GP4 had no effect on FL12 replication. In fact, infectious clones possessing deletions in individual N-glycosylation sites showed increased virus yields compared to the parent FL12 virus. The enhancement of virus replication in GP4 mutated viruses could not be explained. For GP5, Ansari et al. [9] identified a requirement for N44 in the replication of FL12 mutant viruses.

As described above, CD163 in cells coprecipitates with GP2 and GP4. For GP2, there is no correlation between the patterns of N-glycosylation, permissiveness of cells for infection, and the ability of GP2 to interact with CD163. For example, the removal of the N184 glycosylation site, which blocked infection, retained the ability to interact with CD163. For GP4, the removal of N37 and N84 glycosylation sites (Figure 3), which flank the putative neutralization epitope, greatly reduced the interaction between GP4 and CD163, with the unglycosylated form exhibiting the greatest reduction in binding.

Additional evidence for the importance of the GP2, 3, and 4 heterotrimer in the interaction of virus with host cells is found in a large body of data supporting GP4 peptide sequence domains as targets for neutralizing antibody. Kimpston-Burkgren et al. [27] constructed a chimeric virus composed of a PRRSV-1 SD01–08 infectious clone backbone combined with ORFs 2–4 from a PRRSV-2 isolate, FL12. The chimeric virus, called SDFL24, was replication-competent in MARC-145 cells and produced viremia following infection of pigs. Vaccination of pigs with SDFL24 followed by challenge with FL12 or SD01–08 showed significantly greater levels of protection in pigs challenged with FL12. This result demonstrates that GP2–4 are antigen sources for homologous protection. This result was supported by high levels of serum virus neutralizing activity in SDFL24 pigs against FL12, but not SD01–08.

PEPSCAN analysis of a PRRSV-1 proteome, Lelystad virus (LV), identified a reactive oligopeptide, 57-GVSAAQEKISFG-68, in GP4 from LV-infected pigs [24], Figure 3). Antibodies prepared against the oligopeptide neutralized the parent virus, LV, but not PRSSV-1 isolates possessing the GP4 oligopeptides, 57-RVTAAQGRIYTR-68, or 57-RTNTTQGKV----68 (underlined amino acids show the amino acid differences). A comparison of several PRRSV-1 isolates shows that the GP4(57–68) region forms a hypervariable domain [24]. Presumably, the hypervariable domain is responsible for antigenic variation in the epitope and eventual escape from neutralization by antibody. Further support for the importance of GP4(57–68) is found in Costers et al. [28], who characterized several neutralizing mAbs prepared against LV. The mAbs reacted with the same GP4(57–68) oligopeptide. Using a phage display approach to screen sera from pigs infected with the Danish PRRSV-1 isolate, 111/92, Oleksiewicz et al. [29] identified a reactive oligopeptide that covered GP4(59–71). For PRRSV-2 isolates, the GP4(56–68) domain possesses a deletion of three amino acids. As illustrated in Figure 3, the GP4(57–68) region is flanked by two N-glycosylation sites, located at N37 and N84, suggest that the flanking sugar residues shield the epitope from neutralizing antibody. Interestingly, the nucleotide sequence coding for the GP4(57–68) epitope is located within the overlapping region between ORF3 and ORF4.

Even though the GP4(56–68) peptide sequence likely forms a major neutralizing epitope, the participation of this peptide sequence in the formation of an interaction with CD163 remains unclear. Recently, Chen et al. [26] utilized a phage display library to identify peptide sequence domains in CD163 that interact with GP4. Selection was based on the ability of phage to bind immobilized recombinant GP4 derived from a PRRSV-2 isolate. The dodecapeptide, WHEYPLVWLSGY (WHE), showed the highest affinity for GP4. The oligopeptide contains the peptide sequence HEYPLV, is similar to the sequence, HRKPRLV (underlined are the identical amino acids), located at the start of pCD163 SRCR5 polypeptide. A modeling computer program—Spring On-line—identified the GP4(23–48) polypeptide as the predicted binding site for the WHE oligopeptide.

## 4. PRRSV Neutralization and Escape from Homologous and broadly Neutralizing Activity through the GP5/M Heterodimer

In previous work, we conducted virus neutralization (VN) assays using a panel of PRRSV-2 strains along with a single PRRSV-1 isolate, LV. The results showed that pigs could be placed into four distinct VN categories or groups [30]. Group 1 pigs possess no detectable VN activity, even after vaccination and challenge. Group 2 pigs possess only VN activity against the virus used for infection. This is the classic description of homologous VN activity. Group 3 pigs neutralize more viruses than the one used for infection, but not all viruses in the panel. This is a good description of “heterologous” VN activity, where the breadth of neutralization extends beyond the virus used for infection. And finally, Group 4 pigs possess broad VN activity in serum, which further extends the capacity of to neutralize a broad range of PRRSV-2 isolates to include PRRSV-1 strains [31,32]. For experimentally infected pigs, only ~1% of pigs possess broadly neutralizing activity at 42 days after infection.

We identify neutralizing epitopes for PRRSV by selecting for viruses that escape neutralization in the presence of sera possessing homologous or broadly neutralizing activity. The selection steps, described in Trible et al. [32], begin with a 24-well plate in which every well contains a different dilution of virus combined with a different dilution neutralizing serum. After one-hour incubation at 37 °C, the entire well contents are transferred to a 24-well plate of confluent MARC-145 cells. After four days of incubation, wells with the highest concentrations of serum resulting in cytopathic effect (CPE) are pooled. For the second step, the concentration of virus is increased by placing the pooled sample on MARC-145 cells in a T-25 culture flask in the absence of anti-PRRSV serum. After three days, the virus is harvested and the antibody selection step repeated on a new 24-well plate. The selection process is complete when all wells show CPE. To identify amino acids changes associated with escape, ORFs 2–7 corresponding to the structural genes are sequenced and the amino acid sequence compared to the parent virus. We prepared neutralization resistant viruses that escaped neutralization by homologous and broadly neutralizing sera. The escape from homologous neutralization was mapped to mutations in a hypervariable region in GP5, called HVR-2 in Figure 4. A deletion of the tyrosine at position 10 of the M protein in KS62 was associated with escape from broadly neutralizing serum. The Tyr-10 deletion was next to the conserved cysteine at position 9, which forms a disulfide bond with Cys-48 in GP5. The importance of the Tyr-10 deletion was confirmed using reverse genetics and an infectious clone to construct a recombinant virus with the same Tyr-10 deletion. The recombinant deletion virus possessed all of the neutralization properties associated with the parent virus possessing the Tyr-10 deletion. Other properties of the Tyr-10 deletion virus included escape from broadly neutralizing sera from pigs infected with other viruses while retaining sensitivity to serum possessing homologous neutralizing activity. These data show that homologous and broadly neutralizing antibodies recognize different epitopes. Moreover, we recognize that possibility that some of the antibody recognition sites may be more conformational in nature.

As illustrated in Figure 4, the 61 amino acid ectodomain region of PRRSV-2 GP5 contains a peptide signal sequence along with two hypervariable regions (HVRs). Mutations in HVR-1 can create additional glycosylation sites and affect the location of the peptide signal cleavage sites [30]. Two conserved N-glycosylation sites downstream of HVR-1 are located at amino acid positions 44 and 51 for PRRSV-2, 46 and 53 for PRRSV-1, and 44 and 51 for LDV. Near the two N-glycosylation sites is a region occupied by a relatively conserved linear epitope. By using a phage display approach, Ostrowski et al. [33] identified a peptide sequence, called Epitope B, which covered amino acids 37 to 45 of GP5. The peptide sequence is found in a larger epitope characterized by Plagemann et al. [34]. In 2004, Plagemann [35] also identified similar epitopes in GP5 of PRRSV-1 and LDV (Figure 4). A second, short hypervariable region, called HVR-2, is located just downstream of the N51 site [36]. The escape of a virus from homologous neutralization is typically explained by the process of antigenic variation in hypervariable regions of GP4 and GP5 [24,30]. Figure 4 presents a proposed model to explain how a conserved linear epitope, such as the one identified in GP5, escapes neutralization by broadly neutralizing serum. A possible explanation would be that mutations in the M protein, and HVR-1 and HVR-2 of GP5 can change epitope orientation in a manner sufficient to prevent antibody accessibility without changing the conserved epitope’s peptide sequence or its function.

GP5 as a principal target for neutralizing antibody, is supported by research conducted using LDV. Similar to PRRSV-1 and PRRSV-2, LDV GP5 (also known as VP3) possesses two conserved glycosylation sites along with a third site at position 35. Naturally occurring strains of LDV lacking the N35 and N45 sites show a tropism for motor neurons. The cytopathic removal of neurons results in paralysis. Interestingly, the N35/N45 deletion viruses are removed in response to the induction of a strong neutralizing antibody response by the mouse. The remaining residual viruses eventually reacquire both glycosylation sites and become persistent [37]. A similar role for N-glycan shielding in GP5 of PRRSV was demonstrated using reverse genetics of an infectious PRRSV-2 cDNA clone [9]. The results showed that viruses lacking N44 were non-viable in culture, indicating a requirement of N-glycosylation for replication. The removal of the N34 and N51 glycosylation sites resulted in the increased sensitivity of recombinant viruses to neutralization by antibody from pigs infected with the parent or recombinant viruses. Faaberg et al. [38] observed enhanced neutralization activity after infection of pigs with naturally occurring variants of VR-2332, which lacked N30/N33 in GP5. Similar to LDV, the deleted N-glycosylation sites were reacquired during infection of pigs [39]. The restoration of N-glycosylation in GP5 resulted in prolonged infection and the resistance of viruses to neutralizing antibody.

## 5. Summary

Stringent biosecurity measures along with the application of effective vaccines provide the best strategy for the control and eventual elimination of PRRS. To date, modified live attenuated viruses provide the greatest levels of protection, but fall short in a variety of areas. Inactivated virus, subunit, and vectored approaches are important options, but their development is stymied by a failure to provide protective immunity. The complex nature of the PRRSV neutralization response remains a serious obstacle to vaccine development. Several neutralizing epitopes have been identified within the PRRS viral proteins. PEPSCAN analysis identified a short peptide sequence in GP4 of PRRSV-1 as a linear epitope recognized by a monoclonal Ab (mAb) with neutralizing activity [40]. Moreover, Vanhee et al. (2011) [41] characterized additional neutralizing epitopes in GP2 and GP3 proteins of PRRSV-1. In another study done by Zhou et al. (2006) [42], monoclonal antibodies against GP3 of PRRSV-2 were used to identify neutralizing epitopes. Antigenic variation in the viral proteins provides a logical rationale for how viruses can escape homologous neutralizing antibody, but antigenic variation in these proteins fails to explain a broadly neutralizing response directed against highly conserved peptide sequences. A conserved oligopeptide in GP5 flanked by glycosylation sites and hypervariable regions presents an “ideal” broadly neutralizing epitope.

Disrupting the interaction between PRRSV and CD163 can be used as an approach for vaccine development. This review describes an apparent conundrum regarding CD163, its corresponding ligand, GP4, and the overall nature of broad virus neutralization, which appears to localize to GP5. The epitope in GP5 is consistent with a broadly neutralizing epitope; however, the molecular basis for the interaction of the peptide sequence with a corresponding receptor on the macrophage remains unclear and raises an important question: Is there an unidentified cell receptor that interacts with the ectodomain of GP5? The preparation of a panel of monoclonal antibodies possessing homologous, heterologous, and broad neutralization activities is needed to address this conundrum.

## Figures and Tables

**Figure 1 vetsci-06-00009-f001:**
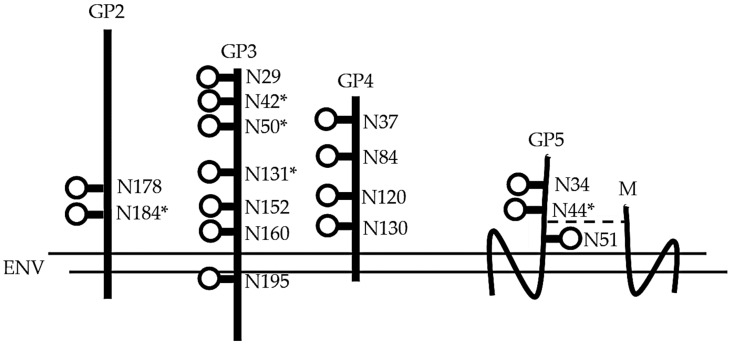
Representation of porcine reproductive and respiratory syndrome virus-2 (PRRSV-2) virion surface proteins. The proteins are shown for a representative PRRSV-2 isolate. The minor glycoproteins GP2–4 form a heterotrimer protruding from the virion surface. The surface is dominated by GP5-M heterodimers. The M protein is nonglycosylated. The position of the glycosylation sites (circles) for GP2–5 are from Das et al. and Ansari et al. [8,9]. Asterisks show those N-sites required for infection [8,9]. The dashed line identifies the disulfide bond between GP5 and M. The structures are not drawn to scale.

**Figure 2 vetsci-06-00009-f002:**
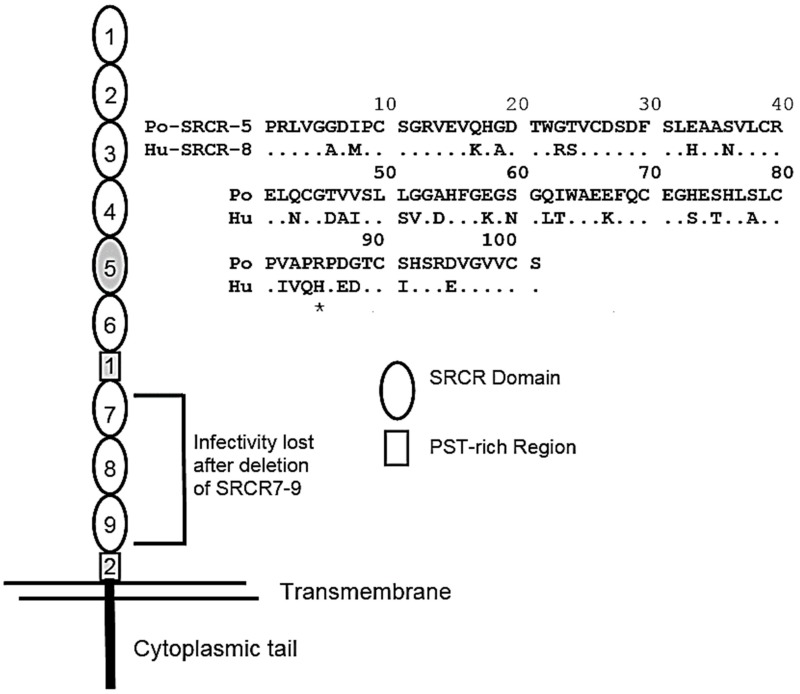
Porcine CD163 organization and SRCR5 peptide sequence. The illustration shows the location of the different domains in porcine CD163 (pCD163). The shaded regions are domains that participate in PRRSV infection of cDNA transfected human embryonic kidney (HEK) cells [19]. The peptide sequences show a comparison between SRCR5 of pCD163 and SRCR8 from the human CD163-like protein [20]. The asterisk shows the location of the R561A mutation described in Ma et al. [21].

**Figure 3 vetsci-06-00009-f003:**
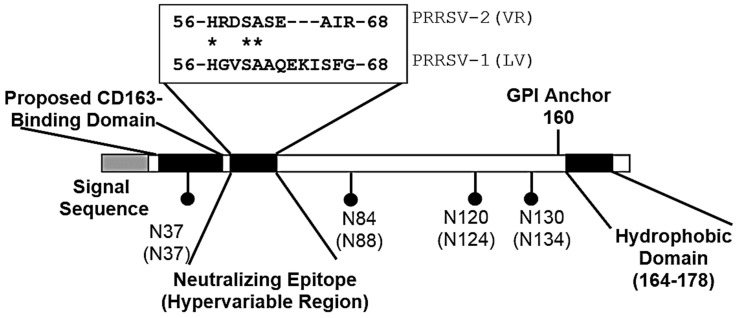
Functional domains in PRRSV GP4. The locations of the N-glycosylation sites are shown for PRRSV-2 (PRRSV-1 in parentheses). The neutralizing epitope—GP4(56–68)—is described in Vanhee et al. [24]. Asterisks show amino acids shared by PRRSV-1 and PRRSV-2. The GPI anchor for PRRSV-2 and the proposed CD163 binding domains are described in Du et al. [25] and Chen et al. [26], respectively. The illustration is based on GP4 from representative PRRSV-1 Lelystad (LV) and PRRSV-2 VR-2332 (VR), GenBank M96262.2 and U87392.3, respectively.

**Figure 4 vetsci-06-00009-f004:**
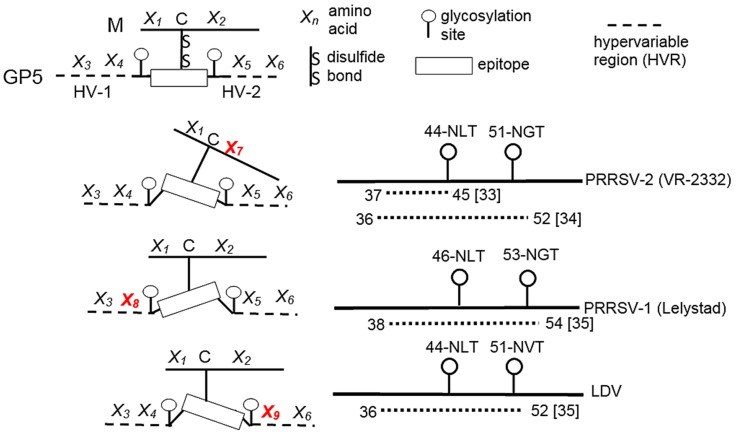
Effect of mutations in GP5 and M proteins on the orientation of a conserved linear epitope. Amino acid mutations (shown in red) influence the orientation of a conserved linear neutralizing epitope. The mutations are sufficient to block neutralizing antibody without affecting how the oligopeptide functions in its interaction with the host cell. N-glycosylation sites provide further shielding. The right-hand figure shows the location of the N-glycosylation sites (circles) and putative neutralizing epitope (dashed lines) for representative PRRS viruses and for lactate dehydrogenase-elevating virus (LDV).

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
