# Peer review of "Challenges for Porcine Reproductive and Respiratory Syndrome (PRRS) Vaccine Design: Reviewing Virus Glycoprotein Interactions with CD163 and Targets of Virus Neutralization"

_vetsci, 2019, doi:10.3390/vetsci6010009_

Round 1

Reviewer 1 Report

The manuscript 'Challenges for porcine reproductive and respiratory syndrome (PRRS) vaccine design: Resolving virus glycoprotein interactions with CD163 and targets of virus neutralization', by Stoian and Rowland revises the current knowledge about the interactions of cell receptors with the surface glycoproteins of PRRSV together with the description of epitopes and the importance of broadly neutralizing epitopes as vaccine candidates .

The paper is generally well written, structured and comprehensible. I enjoyed reading it and, in my opinion, can be accepted for publication in Veterinary Sciences.

Below just a couple of minor comments:

Page 1, line 3; Title: As long as this is a review paper the concept 'resolving' in the title seems inadequate, and i would suggest changing it by 'reviewing'

Page 8, line 298; Summary: I do not agree with the statement that 'The application of effective vaccines provides the best strategy for the control and eventual elimination of PRRS.' The best strategy for the control and eventual elimination of PRRS certainly includes the application of effective vaccines, as much as biosecurity measures to prevent the spread of the disease. Please rephrase the sentence to consider this point.

Page8, line 308; Summary: Again, the statement that 'Disrupting the interaction between PRRSV and CD163 is the logical target for vaccine development' looks too forceful to me. This is one of the options, maybe not the best, certainly not the single one, to tackle vaccine development in PRRSV. The strength of the affirmation should be lowered.

Figures 2, 3 and 4: The quality of the images is not the best, if possible improve the image resolution.

Author Response

Reviewer 1:

Comments and Suggestions for Authors:

The manuscript 'Challenges for porcine reproductive and respiratory syndrome (PRRS) vaccine design: Resolving virus glycoprotein interactions with CD163 and targets of virus neutralization', by Stoian and Rowland revises the current knowledge about the interactions of cell receptors with the surface glycoproteins of PRRSV together with the description of epitopes and the importance of broadly neutralizing epitopes as vaccine candidates .

The paper is generally well written, structured and comprehensible. I enjoyed reading it and, in my opinion, can be accepted for publication in Veterinary Sciences.

Below just a couple of minor comments:

Page 1, line 3; Title: As long as this is a review paper the concept 'resolving' in the title seems inadequate, and i would suggest changing it by 'reviewing'.

Author: This change has been made.

Page 8, line 298; Summary: I do not agree with the statement that 'The application of effective vaccines provides the best strategy for the control and eventual elimination of PRRS.' The best strategy for the control and eventual elimination of PRRS certainly includes the application of effective vaccines, as much as biosecurity measures to prevent the spread of the disease. Please rephrase the sentence to consider this point.

Author: The sentence has been rephrased.

Page8, line 308; Summary: Again, the statement that 'Disrupting the interaction between PRRSV and CD163 is the logical target for vaccine development' looks too forceful to me. This is one of the options, maybe not the best, certainly not the single one, to tackle vaccine development in PRRSV. The strength of the affirmation should be lowered.

Author: This change has been made.

Figures 2, 3 and 4: The quality of the images is not the best, if possible improve the image resolution.

Author: This has been completed.

Reviewer 2 Report

The manuscript “Challenges for porcine reproductive and respiratory syndrome (PRRS) vaccine design: Resolving virus  glycoprotein interactions with CD163 and targets of virus neutralization” by Stoian y Rowland reviews what is known about the interactions between PRRSV glycoproteins and the potential receptors of the virus in the macrophages, mainly CD163. The review is well written and includes most of what is known about PRRSV receptors, their role in the infection cycle and their interaction with viral proteins. Although the information provided is not new and there are some reviews on this matter, the manuscript is interesting and could be published, provided that it is reviewed and changed taking into consideration some comments and suggestions.

First of all, the manuscript focuses on PRRSV glycoproteins and their interactions with target cell receptors and nothing is mentioned in the manuscript about pros and cons of different types of vaccines, strategies for the development of new, improved vaccines or even the importance of neutralizing antibodies for protection. Thus, the title of the manuscript should be changed to remove any reference to vaccines or vaccine development because it could be misleading. In the same way, the summary at the end of the manuscript should be rewritten to avoid any reference to issues which have not been addressed in the main text. Besides, the references to PRDC in the abstract should also be removed because, once again, this issue is not addressed in this review.

Secondly, subheading 4 (i.e. PRRSV neutralization and GP5/M) should be removed or completely rewritten to focus in this matter because in its present form the content of this subheading does not refer to GP5 and M proteins or their role in the development of neutralizing antibodies. Even more, the information provided of this subheading is not relevant for the issues addressed in the manuscript and besides it is not consistent with the rest of the text and breaks the line of thinking when reading the manuscript.

In the same way subheading 5 (i.e. Escape from homologous and broadly neutralizing activity locates to the GP5/M heterodimer) should be mostly rewritten and rephrased to make it more precise, to better adjust it to what is known about this issue and to avoid speculation. To star with, part of the information (that related to the description of proposed neutralizing epitopes in GP5 and M proteins) would better fix within the previous subheading. Besides, in the text it is suggested that escape to homologous neutralization is due to mutations in GP5 and escape to heterologous neutralization is due to a deletion in protein M. However, these seem to be the results of one particular study and might not fully represent or explain the complex picture of PRRSV homologous and heterologous neutralization. On one hand, the question arise of whether only these two changes (i.e. mutation in GP5 and deletion in protein M) were observed in the neutralization escape mutants analyzed in that study. Were there not other changes in these or other structural proteins? Were the described changes consistent and found in all mutants analyzed? Why pooling positive well? (it could select some mutants and abrogate others). On the other hand, the accepted fact that PRRSV receptor CD163 interacts with GP2 and GP4 seems to be contradictory with the idea that GP5 and M are the main responsible for the generation of neutralizing antibodies both strain specific and broadly reactive. How do the authors explain this apparent contradiction? Are they proposing that neutralizing epitopes identified in other proteins are irrelevant in terms of homologous and heterologous neutralization? This discussion and the role of other neutralizing epitopes should be included in the text. In addition to the abovementioned issues, it should be taken into consideration that the role and even de existence of the neutralizing epitope described in the ectodomain of GP5 has been questioned for European isolates (Vanhee et al., 2011. Characterization of antigenic regions in the porcine reproductive and respiratory syndrome virus by the use of peptide-specific serum antibodies. Vaccine, 29: 4794-4804). This fact should also be included in the text and discussed. Furthermore, other linear neutralizing epitopes have been described in other PRRSV proteins, including GP2, GP3 and GP4 which might also be responsible for the development of neutralizing antibodies implicated in the neutralization of the homologous or heterologous strains. This phenomenon has been demonstrated in the case of PRRSV 1 for GP4, which contains a very variable neutralizing epitope which seem to be one of the epitopes responsible for the development of strain specific neutralizing antibodies and for the emergence of neutralization escape mutants during the time course of infection (Costers et al., 2010. GP4-specific neutralizing antibodies might be a driving force in PRRSV evolution. Virus Res, 154:104-113). Again, this information should be included in the text and discussed. Finally, the possibility of the existence of conformational neutralizing epitopes which might be, at least partially, responsible for the development of neutralizing antibodies should be discussed because these epitopes might exist, particularly taking into consideration that it is a heterotrimer and not a single protein the one interacting with the main PRRSV receptor CD163.

On the other hand, it should be clearly stated that it has not been demonstrated that changes in GP5 epitope orientation due to mutations in the surrounding areas of GP5 or in the ectodomain of protein M can prevent antibody accessibility and that this is just a theory. Thus, the sentence “Mutations in the M protein, and HVR-1 and HVR-2 of GP5 can change epitope orientation in a manner sufficient to prevent antibody accessibility without changing the conserved epitope’s peptide sequence or its function” (lines 278-280) should be rephrased to better indicate that this is a possibility and a theory which has not be proven.

In addition, particularities of PRRSV-1 and PRRSV-2, which exist, should be indicated.

Other comments and modifications include:

Although it is true that Kuhn et al. proposed in 2016 a new genus, named Rodarterivirus the International Committee on Taxonomy of Viruses has created the genus Betaarterivirus within the family Arteriviridae with three subgenus, two of which (i.e. Ampobartevirus and Eurpobartevirus) include the two species created for the original PRRSV-1 and PRRSV-2 (i.e. Betaarterivurs suid 1 and Betaarterivurs suid 2, respectively). This should be indicated in the text because the genus Rodarterivirus has never been created.

The sentence “A comparison of several PRRSV-1 isolates shows that the GP4 (57-68) region forms a hypervariable domain” (lines 192-193) should be referred.

The sentence “For PRRSV-2 isolates, the GP4 (56-68) domain possesses a deletion of 3 amino acids, which explain the absence of cross-neutralization between PRRSV-1 and PRRSV-2” (lines 199-200) should be deleted because it is speculative and misleading. On one hand, many changes are observed in the amino acid sequence of the neutralizing epitope described GP4 both among European viruses and between European and American viruses. These changes might explain the lack of cross-reactivity independently of the existence or not of a deletion in this portion of the genome. On the other hand, other neutralizing epitopes have been described in other structural proteins which might be totally or partially responsible for the lack of cross-reactivity between European and American viruses. Finally, and more important, the actual role of the described deletion has not been studied and proven.

Sentence “This demonstrates that the overlapping nucleotides shared between the PRRSV ORFs do not induce serious constraints on peptide sequence variability” (lines 204-205) should be deleted because it might not be true for all proteins.

Sentence “The cytopathic removal of neutrons results in results in in paralysis” (lines 284-285) makes no sense and should be deleted and reformulated.

Neutrons should be changed by neurons in line 284 (i.e. “…the N35 and N45 sites show a tropism for motor neutrons).

Author Response

Reviewer 2:

Comments and Suggestions for Authors:

The manuscript “Challenges for porcine reproductive and respiratory syndrome (PRRS) vaccine design: Resolving virus glycoprotein interactions with CD163 and targets of virus neutralization” by Stoian y Rowland reviews what is known about the interactions between PRRSV glycoproteins and the potential receptors of the virus in the macrophages, mainly CD163. The review is well written and includes most of what is known about PRRSV receptors, their role in the infection cycle and their interaction with viral proteins. Although the information provided is not new and there are some reviews on this matter, the manuscript is interesting and could be published, provided that it is reviewed and changed taking into consideration some comments and suggestions.

First of all, the manuscript focuses on PRRSV glycoproteins and their interactions with target cell receptors and nothing is mentioned in the manuscript about pros and cons of different types of vaccines, strategies for the development of new, improved vaccines or even the importance of neutralizing antibodies for protection. Thus, the title of the manuscript should be changed to remove any reference to vaccines or vaccine development because it could be misleading. In the same way, the summary at the end of the manuscript should be rewritten to avoid any reference to issues which have not been addressed in the main text. Besides, the references to PRDC in the abstract should also be removed because, once again, this issue is not addressed in this review.

Author: The reference to PRDC was removed from the second sentence. Although we do not mention in the manuscript any pros and cons of different types of vaccines or strategies for the development of new, improved vaccines, we do present an important view on why the current vaccines are inefficient – even though GP2 and GP4 seem to be the viral proteins interacting with CD163, the escape from broadly neutralizing antibodies locates to GP5/M. Therefore, we believe that an effective vaccine needs to address both aspects.

Secondly, subheading 4 (i.e. PRRSV neutralization and GP5/M) should be removed or completely rewritten to focus in this matter because in its present form the content of this subheading does not refer to GP5 and M proteins or their role in the development of neutralizing antibodies. Even more, the information provided of this subheading is not relevant for the issues addressed in the manuscript and besides it is not consistent with the rest of the text and breaks the line of thinking when reading the manuscript.

Author: Subheading 4 was shortened and incorporated into subheading 5.

In the same way subheading 5 (i.e. Escape from homologous and broadly neutralizing activity locates to the GP5/M heterodimer) should be mostly rewritten and rephrased to make it more precise, to better adjust it to what is known about this issue and to avoid speculation. To star with, part of the information (that related to the description of proposed neutralizing epitopes in GP5 and M proteins) would better fix within the previous subheading. Besides, in the text it is suggested that escape to homologous neutralization is due to mutations in GP5 and escape to heterologous neutralization is due to a deletion in protein M. However, these seem to be the results of one particular study and might not fully represent or explain the complex picture of PRRSV homologous and heterologous neutralization. On one hand, the question arise of whether only these two changes (i.e. mutation in GP5 and deletion in protein M) were observed in the neutralization escape mutants analyzed in that study. Were there not other changes in these or other structural proteins? Were the described changes consistent and found in all mutants analyzed? Why pooling positive well? (it could select some mutants and abrogate others). On the other hand, the accepted fact that PRRSV receptor CD163 interacts with GP2 and GP4 seems to be contradictory with the idea that GP5 and M are the main responsible for the generation of neutralizing antibodies both strain specific and broadly reactive. How do the authors explain this apparent contradiction? Are they proposing that neutralizing epitopes identified in other proteins are irrelevant in terms of homologous and heterologous neutralization? This discussion and the role of other neutralizing epitopes should be included in the text. In addition to the abovementioned issues, it should be taken into consideration that the role and even de existence of the neutralizing epitope described in the ectodomain of GP5 has been questioned for European isolates (Vanhee et al., 2011. Characterization of antigenic regions in the porcine reproductive and respiratory syndrome virus by the use of peptide-specific serum antibodies. Vaccine, 29: 4794-4804). This fact should also be included in the text and discussed. Furthermore, other linear neutralizing epitopes have been described in other PRRSV proteins, including GP2, GP3 and GP4 which might also be responsible for the development of neutralizing antibodies implicated in the neutralization of the homologous or heterologous strains. This phenomenon has been demonstrated in the case of PRRSV 1 for GP4, which contains a very variable neutralizing epitope which seem to be one of the epitopes responsible for the development of strain specific neutralizing antibodies and for the emergence of neutralization escape mutants during the time course of infection (Costers et al., 2010. GP4-specific neutralizing antibodies might be a driving force in PRRSV evolution. Virus Res, 154:104-113). Again, this information should be included in the text and discussed. Finally, the possibility of the existence of conformational neutralizing epitopes which might be, at least partially, responsible for the development of neutralizing antibodies should be discussed because these epitopes might exist, particularly taking into consideration that it is a heterotrimer and not a single protein the one interacting with the main PRRSV receptor CD163.

Author: For this study, seven isolates representing the genetic diversity of PRRSV were selected. ORF2 2-7 were sequenced and the only mutated regions were located in GP5 and M protein. Only the positive wells for CPE were pooled because these would show escape from neutralization in presence of neutralizing sera. More details regarding the GP2-4 neutralizing epitopes identified in different papers were included in the summary subheading. Nonetheless, the scope of this review was to identify a potential cause for the failure of the current vaccines – although several neutralizing  epitopes have been identified in GP2-4, the presence of a broadly neutralizing epitope in GP5 needs to be taken into account.

On the other hand, it should be clearly stated that it has not been demonstrated that changes in GP5 epitope orientation due to mutations in the surrounding areas of GP5 or in the ectodomain of protein M can prevent antibody accessibility and that this is just a theory. Thus, the sentence “Mutations in the M protein, and HVR-1 and HVR-2 of GP5 can change epitope orientation in a manner sufficient to prevent antibody accessibility without changing the conserved epitope’s peptide sequence or its function” (lines 278-280) should be rephrased to better indicate that this is a possibility and a theory which has not be proven.

Author: This has been completed.

In addition, particularities of PRRSV-1 and PRRSV-2, which exist, should be indicated.

Author: The presence of different neutralizing epitopes in GPs2-4 of PRRSV-1 and PRRSV-2 was included in the text.

Other comments and modifications include:

Although it is true that Kuhn et al. proposed in 2016 a new genus, named Rodarterivirus the International Committee on Taxonomy of Viruses has created the genus Betaarterivirus within the family Arteriviridae with three subgenus, two of which (i.e. Ampobartevirus and Eurpobartevirus) include the two species created for the original PRRSV-1 and PRRSV-2 (i.e. Betaarterivurs suid 1 and Betaarterivurs suid 2, respectively). This should be indicated in the text because the genus Rodarterivirus has never been created.

Author: This has been completed.

The sentence “A comparison of several PRRSV-1 isolates shows that the GP4 (57-68) region forms a hypervariable domain” (lines 192-193) should be referred.

Author: The reference was added to the text.

The sentence “For PRRSV-2 isolates, the GP4 (56-68) domain possesses a deletion of 3 amino acids, which explain the absence of cross-neutralization between PRRSV-1 and PRRSV-2” (lines 199-200) should be deleted because it is speculative and misleading. On one hand, many changes are observed in the amino acid sequence of the neutralizing epitope described GP4 both among European viruses and between European and American viruses. These changes might explain the lack of cross-reactivity independently of the existence or not of a deletion in this portion of the genome. On the other hand, other neutralizing epitopes have been described in other structural proteins which might be totally or partially responsible for the lack of cross-reactivity between European and American viruses. Finally, and more important, the actual role of the described deletion has not been studied and proven.

Author: This change has been made.

Sentence “This demonstrates that the overlapping nucleotides shared between the PRRSV ORFs do not induce serious constraints on peptide sequence variability” (lines 204-205) should be deleted because it might not be true for all proteins.

Author: This has been completed.

Sentence “The cytopathic removal of neutrons results in results in in paralysis” (lines 284-285) makes no sense and should be deleted and reformulated.

Author: This has been completed.

Neutrons should be changed by neurons in line 284 (i.e. “…the N35 and N45 sites show a tropism for motor neutrons).

Author: This has been completed.